# Functional, Nutritional, and Sensory Quality of Mixed Flours-Based Breads as Compared to Durum Wheat Semolina-Based Breads

**DOI:** 10.3390/foods10071613

**Published:** 2021-07-12

**Authors:** Mariagrazia Molfetta, Giuseppe Celano, Fabio Minervini

**Affiliations:** Dipartimento di Scienze del Suolo, della Pianta e degli Alimenti, Università degli Studi di Bari Aldo Moro, Via Amendola 165/A, 70126 Bari, Italy; mariagrazia.molfetta@gmail.com (M.M.); fabio.minervini@uniba.it (F.M.)

**Keywords:** bread, glycaemic index, antioxidant activity, essential free amino acids, gamma-aminobutyric acid

## Abstract

Increasing preference of consumers and bakers towards bread manufactured with mixed flours and/or sourdough drove us to investigate about influence of flours and sourdough on crumb grain, chemical, sensory, and in vitro glycaemic index (GI) and antioxidant activity of bread. To this aim, we produced and compared six experimental breads: three were based on a mixture of flours (soft wheat, durum wheat semolina, barley, oat, rye, and buckwheat); three were semolina-based breads. Two different sourdoughs (wheat or mixed flours) were assessed. Compared to semolina breads, those containing a mixture of flours showed higher specific volume. The use of sourdough led to increased concentrations of total free amino acids (FAA). Mixed flours bread with addition of mixed flours sourdough was rich in some essential FAA and amino acid derivative bioactive gamma-aminobutyric acid. Type of flours had higher influence than sourdough addition on volatile organic compounds. All the mixed flours breads, although showing profiles of volatile organic compounds differing from those of semolina breads, resulted acceptable. In addition, they had lower GI and higher antioxidant activity than semolina breads. Type of flours had much higher impact on GI and antioxidant activity than sourdough.

## 1. Introduction

In many countries, bread represents, at the same time, one of the pillars of cultural heritage and the major source of dietary carbohydrates [1]. Refined flour from soft wheat (*Triticum aestivum*) is the most used flour for bread-making [2], allowing to obtain the so-called “white wheat bread”, whose sensory traits are highly appreciated by most of consumers [1]. However, white wheat bread contains lower concentration of micronutrients and dietary fiber than bread based, at least in part, on wholemeal flour [3]. For this reason, as well as for correlation between refined-grain foods and increased risk of metabolic syndrome and type 2 diabetes, white wheat bread has been sitting on the dock for decades [4]. Although durum wheat (*Triticum turgidum* subsp. *durum*) generally has lower bread-making performances than soft wheat, breads based on durum wheat semolina (e.g., Pagnotta del Dittaino PDO, Pane di Altamura PDO, Pane di Matera PGI) are very popular in some regions (e.g., southern Italy) of the Mediterranean area [5]. Besides wheat, rye (*Secale cereale*) flour is largely used for bread-making, especially in northern, eastern, and central Europe [6]. Among minor cereals, barley (*Hordeum vulgare*) and oat (*Avena sativa*) flours have been ever employed for manufacturing bread [7,8], either in combination with wheat or, more seldom (due to their poor bread-making aptitude), alone [9,10]. Besides cereals, other crops have been traditionally used for obtaining flours as basic ingredients of bread. Among them, buckwheat (*Fagopyrum esculentum*) is a pseudo-cereal, whose flour can be used [11], although the resulting bread, compared to wheat bread, has worse structure, darker color [12] and bitter taste [13]. Compared to refined soft wheat flour, durum wheat, rye, barley, oat, and buckwheat give flours with higher concentrations of dietary fiber, essential amino acids, and phytochemicals [14].

Although baker’s yeast is still the most used starter in modern bread-making, the use of sourdough, as either leavening agent or baking improver, attracts an increasing number of bakers, allowing to respond to consumers’ demand for tasteful leavened baked goods, with longer shelf-life [15]. In addition, sourdough lactic acid bacteria and yeasts contribute to improve several nutritional (e.g., concentration of essential free amino acids) and functional (e.g., concentration of bioactive compounds, such as gamma-aminobutyric acid) features of bread [16].

Perception of white wheat bread as a scarcely healthy food item is one of the drivers causing a decrease of bread consumption in Europe in the past decades [17]. However, since the first decade of the new millennium, consumers are increasingly preferring breads other than white wheat bread, such as those (labeled as “multi-cereal”, “multi-grain”, or “mixed flours”) based on a mixture of flours from cereals and pseudo-cereals [18]. More and more bakers are following this trend, proposing mixed flours-based breads, whose unusual sensory features and claimed beneficial health effects are appealing to consumers. However, replacement of refined soft wheat flour with flours with inferior bread-making aptitude poses challenges to bakers in terms of bread texture, taste, and overall acceptability [19].

This study aimed to assess the influence of a mixture of flours (wheat, barley, oat, rye, buckwheat) traditionally used in Europe for bread-making, as well as the influence of sourdough as baking improver on chemical, sensory features, and in vitro glycaemic index and antioxidant activity of bread. The experimental plan included the comparison of mixed flours-breads with those based on durum wheat semolina, and the use of baker’s yeast, as the main leavening agent.

## 2. Materials and Methods

### 2.1. Dough Ingredients and Sourdough Preparation

The following flours were purchased from local markets and used for sourdough preparation and bread-making: durum wheat semolina (Casillo s.r.l., Corato, Italy), refined (type 00) soft wheat (deformation energy index, W = 170 × 10^−4^ J, Casillo), soft wheat (type 00, W = 330 × 10^−4^ J, Molino Rossetto SpA, Pontelongo, Italy), wholemeal rye (Molino Spadoni S.p.A., Coccolia, Italy), wholemeal barley (Sottolestelle srl, San Giovanni Rotondo, FG, Italy), oat (Molino Rossetto), and wholemeal buckwheat (Le Farine Magiche, Frigento, AV, Italy). Gross chemical composition of flours, as declared on the labels of flour packages, is shown in Appendix A.

Mixed flours sourdough (MFS) was obtained upon spontaneous fermentation followed by daily continuous back-slopping steps. In detail, the production of sourdough began with mixing (for 5 min) 25 g of soft wheat flour (W = 170 × 10^−4^ J), 12.5 g of durum wheat semolina, 9.375 g of rye flour, 6.25 g of barley flour, 6.25 g of oat flour, 3.125 g of buckwheat flour with 37.5 g of tap water. The dough (total weight: 100 g; dough yield: 160) was fermented for 16 h at 30 °C. After a resting step (8 h at 4 °C), 10 g of the fermented dough were used as inoculum (“back-slopping” step) of a new dough (100 g), composed of 56.25 g of mixed flours and 33.75 g of water. This new dough was fermented for 16 h at 30 °C and, after the resting step (8 h at 4 °C), used as inoculum for the subsequent back-slopping. The back-slopping, fermentation, and resting steps were daily performed until a mature sourdough (constant pH and volume increase after fermentation) was obtained. pH and volume of the doughs were measured daily at the beginning and at the end of fermentation. Microbiological analyses were performed on the fermented doughs at 10th and 20th day. In parallel with the MFS, a wheat sourdough, kindly given by Francesco Colella (Spacciagrani, Conversano, Italy), was daily propagated, through back-slopping. This 25-year-old wheat sourdough was obtained using a traditional protocol, consisting of spontaneous fermentation of wheat flour and water, followed by continuous back-slopping and fermentation steps. We will refer to this sourdough as “traditional wheat sourdough” (TWS). Propagation of TWS envisaged the use of 50 g of refined soft wheat flour (type 0, Mulino Marino, Cossano Belbo, CN, Italy), mixed with 50 g of TWS, and 25 g of tap water. This dough was fermented for 3 h at 30 °C (final pH = 3.9 ± 0.03) before being used as bread ingredient.

### 2.2. Enumeration of Lactic Acid Bacteria and Yeasts

Cell densities of presumptive lactic acid bacteria were enumerated using modified (5 g/L maltose, 50 mL/L fresh yeast extract, pH 5.6) MRS (Oxoid Ltd., Basingstoke, UK) agar medium supplemented with cycloheximide (0.1 g/L). Yeasts were enumerated using Sabouraud Dextrose agar medium (Oxoid) supplemented with chloramphenicol (0.1 g/L). Before inoculating the media, sourdoughs were homogenized with 90 mL of sterile saline solution (NaCl 9 g/L), serially diluted and pour-plated. Colonies were counted after 48 h of incubation at 30 °C.

### 2.3. Bread-Making

As mixed flours-based sourdough reached maturity, bread-making was performed at the pilot plant of the Department of Soil, Plant and Food Sciences of the University of Bari. Six breads were produced: durum wheat semolina bread with traditional wheat sourdough (S-TWS); durum wheat semolina bread with mixed flours sourdough (S-MFS); mixed flours bread started just with baker’s yeast (MF); mixed flours bread with traditional wheat sourdough (MF-TWS); mixed flours bread with mixed flours sourdough (MF-MFS); durum wheat semolina bread started just with baker’s yeast (S) (Figure 1).

Bread-making protocol (percentage of ingredients, time and temperature of fermentation, time of baking) had been preliminary set-up. In detail, percentage of water was defined empirically, adding the amount of water needed to get a dough having the same firm consistency for all the breads. For mixed-flours breads, percentages of each flour were defined based on capacity of reaching a minimal value (2.5 cm^3^/g) for specific volume. The aim was to limit, as much as possible, the percentage of soft wheat flour, without worsening bread-making performances. Ingredients of breads, except for baker’s yeast (0.5%) and salt (1%), which were ever used, are detailed in Table 1.

The bread-making protocol consisted of two steps: (i) doughs, consisting of either semolina or mixed flours (67.5 g), baker’s yeast (1 g), and tap water (67.5 g), were fermented for 16 h at 25 °C; (ii) pre-fermented doughs were added with either semolina or mixed flours (45 g), salt (2 g) and, except for MF and S breads, sourdough (20 g), and mixed for 5 min with a planetary mixer (Electrolux Orbital stand mixer, model EKM4000). We used baker’s yeast also for breads added with sourdough because it guarantees dough leavening in short time (during the second step). After fermentation (1.5 h at 30 °C), breads were baked at 220 °C for 20 min, in a rotary oven (Combo 3, Zucchelli, Verona, Italy) (Figure 1).

### 2.4. Crumb Grain Characteristics of Breads

The specific volume of the loaves was determined using the rapeseed displacement method AACC 10-05.01 [20]. To analyze gas cells, breads were cut into slices and scanned. Images were transformed from a color scale into a grey scale and analyzed using the UTHSCSA ImageTool 3.0 freeware program. Grey scale images were transformed into binary (white or black pixels) images, giving to all images the same value of the parameter “threshold” (T = 130). The lower the threshold is, the lower the number of gas cells (interpreted as black pixels) is recognized by the software. Gas cells number in bread crumb was expressed as percentage ratio between area filled by gas cells and the total area [21].

### 2.5. Chemical Characteristics of Breads

pH value and total titratable acidity (TTA) of breads were determined after homogenization of 10 g of bread with 50 mL of distilled water. pH was measured with a Crison 507 pH-meter (Crison, Milan, Italy). TTA was expressed as the milliliters of NaOH (4 g/L) required to get a pH of 8.3. PH s.r.l. laboratories (Barberino Tavernelle, Italy) analyzed breads for concentrations of ash, moisture, total fat, total saturated fatty acids [22], total carbohydrates, sugars (galactose, glucose, fructose, sucrose, lactose, maltose) composition (internal protocol), proteins (internal protocol), and salt (internal protocol). The determination of soluble dietary fiber and total dietary fiber in breads was carried out by PH s.r.l. laboratories, following the approved method AOAC 993.19 [23] and ISTISAN Reports 1996/34 [22], respectively.

Water-soluble extracts of breads were prepared [24] and used to determine concentration of organic acids and free amino acids (FAA). In detail, preliminarily the dry weight of breads was evaluated using a thermogravimetric balance (MA35, Sartorius Stedim Biotech GmbH, Germany). Four grams of bread were suspended with Tris-HCl (6.057 g/L) buffer pH 8.8 (volume ranging from 10 to 12 mL according to the dry weight of each sample). The suspension was incubated (4 °C, 1 h), vortexed every 15 min, and finally centrifuged (10,000× *g*, 20 min). One volume of the supernatant was mixed with one volume of perchloric acid (0.5 g/L) and incubated overnight at 4 °C. After incubation, the mix was centrifuged (10,000× *g*, 10 min). The supernatant was recovered, filtered (0.20 μm), and subjected to High Performance Liquid Chromatography (HPLC) analysis. An ÄKTA Purifier system (GE Healthcare Bio-Sciences, Uppsala, Sweden), equipped with an Aminex HPX-87H column (Bio-Rad Laboratories, Inc., Hercules, CA, USA) and UV detector operating at 210 nm, was used. Organic acids were eluted isocratically with H_2_SO_4_ (0.981 g/L), at a flow rate of 0.6 mL/min, with the column set at 60 °C. Concentrations of lactic and acetic acids in bread were calculated using external standards [21].

Water-soluble extracts of bread, after proteins precipitation, were subjected to determination of FAA through the Amino Acid Analyzer Biochrom 30+ (Biochrom Ltd., Cambridge Science Park, UK), equipped with a Lithium High Performance Physiological Column (Biochrom Ltd., Cambridge Science Park, England). Elution and derivatization of FAA were carried out following the protocol indicated by the manufacturer. Individual FAA were quantified using the external standard method, after having analyzed amino acid standard (AAS18 Supelco, Merck KGaA, Darmstadt, Germany) solutions, supplemented with gamma-aminobutyric acid (GABA), ornithine, and tryptophan, at known concentrations [25].

### 2.6. Profile of Volatile Organic Compounds

To detect volatile organic compounds (VOCs) of bread samples, gas chromatography/mass spectrometry (GC/MS) analyses were performed after extraction of VOCs by headspace solid-phase micro-extraction (HS-SPME) sampling technique. First, 0.75 g of ground bread were placed in 20 mL glass vials and 10 μL of internal standard solution (4-methyl-2-pentanol, 0.033 g/L) were added. A COMBIPAL-xt autosampler (CTC Analysis AG, Zwingen, Switzerland) was used to standardize the extraction procedure. Bread samples were equilibrated at 60 °C for 10 min. After equilibration, a divinylbenzene/carboxen/polydimethylsiloxane fiber (Supelco, Bellefonte, PA, USA) was exposed to headspace for 50 min at 60 °C. The extracted VOCs were subjected to 3 min-long desorption in splitless mode at 230 °C and then injected into a Clarus 680 (Perkin Elmer, Waltham, MA, USA) gas-chromatography system equipped with a Rtx-WAX (30 m × 0.25 mm i.d., 0.25 μm film thickness) capillary column (Restek Superchrom, Milan, Italy), using the temperature diagram and carrier gas conditions reported by Vitellio et al. [26]. A single-quadrupole mass spectrometer Clarus SQ8MS (Perkin Elmer), coupled to the gas-chromatography system, was used to detect the VOCs. Peaks were identified and VOCs were quantified as reported previously [26].

### 2.7. Prediction of Glycaemic Index

Glycaemic index (GI) of bread was predicted upon determination of starch hydrolysis index (HI). In vivo digestion of starch contained in bread samples was simulated [27]. Starch content was estimated through enzymatic determination (D-Fructose/D-Glucose assay kit, Megazyme Intl., Ireland) of glucose concentration. Glucose was converted (factor = 0.9) into starch concentration. Aliquots of bread, containing one gram of starch, were given to ten volunteers for obtaining boluses. Subsequently, boluses were subjected to in vitro sequential digestion with pepsin-HCl and pancreatic amylase. Simulated digests were dialyzed (cut-off of the membrane: 12,400 Da) for 180 min. Aliquots of dialysate, containing free glucose and partially hydrolyzed starch, were sampled every 30 min and further treated with amylo-glucosidase. Then, free glucose was determined using the above-mentioned enzyme-based kit and finally converted into hydrolyzed (digested) starch in breads. Starch HI was calculated as percentage ratio between the area under the curve (0–180 min) obtained for experimental bread and the corresponding area obtained for a commercial white wheat bread chewed by the same volunteer. The equation GI = 0.549 HI + 39.71 was used for the prediction of GI [28].

### 2.8. Prediction of Antioxidant Activity

The radical scavenging activity was determined on the methanolic extract (ME) of breads [29]. Three grams of ground bread were mixed with 30 mL of 80% (vol/vol) methanol to get ME, which was purged from oxygen by stirring the suspension for 60 min. After centrifugation (4600× *g*, 20 min, 4 °C), the supernatant (ME) was collected and tested for scavenging activity using 2,2-DiPhenyl-1-PicrylHydrazyl (DPPH•) as free radical. A solution of DPPH• (0.246 g/L) in methanol (80%) was freshly prepared. Reaction mixture, consisting of 167 μL DPPH•, 167 μL of ME, and 667 μL of methanol (80%), was incubated (20 °C, protected from light). The absorbance of the mixture (A_sample_) was read at 517 nm after 10 min of incubation. Butylhydroxytoluene (0.45 g/L in methanol 80%) was used as positive control. Absorbance of a blank, consisting of 167 μL DPPH• and 833 μL of methanol (80%), was read (at a wavelength of 517 nm) at the beginning of incubation and after 10 min. Values of absorbance of the blank after 0 and 10 min were averaged (A_Blank_) and used as reference for calculating the antioxidant activity, after 10 min, as follows:DPPH• radical scavenging activity = [(A_Blank_ − A_sample_)/A_Blank_] × 100(1)

### 2.9. Sensory Analysis

Breads were subjected to quantitative descriptive analysis after 3 h from baking. The panel was composed of ten volunteers (5 female and 5 male, range 25–43 years old) from laboratory staff, who were previously trained about the meaning of the sensory attributes and scores. Before the panel test, performed in a classroom, breads were cut into slices (1.5 cm thick), which were divided into five pieces and served at 22 °C and under normal lighting. Breads were identified by unique blinding numerical codes and served in randomized complete block order. Each panellist evaluated two pieces of bread per thesis (total number of samples: 12) and rinsed her/his mouth with still water (at 20 °C) between testing two consecutive pieces of bread. The attributes were: crust color, odor, crispness, elasticity, masticability, acid taste, sweet taste, salty taste, aroma, and overall acceptability. The score for each sensory attribute ranged from 1 (lowest) to 5 (highest) [30].

### 2.10. Statistical Analyses

Data (three biological replicates, analyzed in duplicate) were subjected to one-way analysis of variance (ANOVA), and pairwise comparison of treatment means was achieved by Tukey’s procedure at *p* < 0.05, using the software Statistica 12 (StatSoft, Inc, Matulsa, OK, USA). In addition, the results of all the analyses (except for sensory analysis) performed on bread were subjected to Principal Components Analysis (PCA) using Statistica 12.

## 3. Results

### 3.1. Main Features of Sourdoughs

Acidification and leavening performance of the mixed flours sourdough (MFS) were not significantly (*p* > 0.05) different after 10 and 20 days of daily propagation (Appendix A). The ratio between cell density of lactic acid bacteria (LAB) and yeasts was 100:1. After 20 days of propagation, MFS, having a pH of 3.81 ± 0.04, was considered as a mature sourdough, and used as baking improver. Leavening performance of traditional wheat sourdough (TWS) did not differ (*p* > 0.05) from that of MFS (data not shown). TWS harbored LAB at a cell density of 8.3 ± 0.2 log cfu/g, lower (*p* < 0.05) than that found in MFS. Yeasts were found in TWS at a cell density of 7.2 ± 0.3 log cfu/g, not significantly (*p* > 0.05) different from yeast cell density in MFS.

### 3.2. Crumb Grain and Chemical Characterization of Breads

All the mixed flours-based breads (MF, MF-TWS, and MF-MFS) showed higher (*p* < 0.05) specific volume than the semolina-based breads (S-TWS, S-MFS, and S), without any significant differences (*p* > 0.05) among them (Table 2). MF-MFS bread was characterized by the highest (*p* < 0.05) percentage of gas cell area, followed by MF and MF-TWS breads. The lowest (*p* < 0.05) gas cell area was found for S-TWS and S breads. Regarding gross chemical composition, fat was higher (*p* < 0.05) in the mixed flours- than in semolina-based breads, whereas fructose and maltose were detected just in the semolina-based breads. No significant differences (*p* > 0.05) were found for all the other chemical components, including soluble and total dietary fiber. Mixed flours-based sourdough breads (MF-TWS and MF-MFS) were characterized by the lowest and highest (*p* < 0.05) values of pH and TTA, respectively. Semolina bread without sourdough (S) had the highest pH (and lowest TTA), whereas the remaining breads were characterized by intermediate acidity parameters. All the mixed flours-based breads contained the highest concentration (*p* < 0.05) of lactic acid. In addition, mixed flours-based bread with addition of MFS (MF-MFS) contained the highest (*p* < 0.05) concentration of acetic acid (Table 2).

Concentration of total FAA greatly varied depending on the bread, ranging from ca. 1500 (S bread) to ca. 4100 (MF-MFS bread) mg/kg (Table 3). Regarding individual FAA, mixed flours-based sourdough breads showed the highest (*p* < 0.05) concentrations of two essential amino acids (isoleucine and lysine), and of glycine, proline, and ornithine (Table 3). In addition, MF-MFS contained the highest (*p* < 0.05) concentrations of aspartate, alanine, tyrosine, histidine, and of the essential amino acids valine, leucine, phenylalanine, and tryptophan, whereas MF-TWS contained serine, glutamate, and cysteine (another essential amino acid) at the highest levels (*p* < 0.05). Furthermore, MF-MFS contained the amino acid derivative gamma-aminobutyric acid (GABA) at the highest (*p* < 0.05) concentration. For most of FAA (e.g., valine, phenylalanine, lysine), the lowest (*p* < 0.05) concentrations were found either in MF or S breads, where no sourdough had been added (Table 3).

### 3.3. Profile of Volatile Organic Compounds in Breads

GC/MS analysis was performed to determine the profiles of VOCs in breads. Seventy-three VOCs were detected and grouped into nine chemical classes (Appendix A). The class of alcohols included 15 compounds, among which phenylethyl alcohol was the most representative, ranging from 5.89 (MF) to 16.53 (S-TWS) mg/kg of bread. Benzyl alcohol (1.06–1.88 mg/kg), ethanol (0.96–2.67 mg/kg), and 1-butanol,3-methyl (0.71–1.46 mg/kg) were other representative alcohols. Among the mixed flours-based breads, no significant difference (*p* > 0.05) of concentrations was found for all the alcohols, except for 1-butanol,3-methyl and 1-hexanol (highest in MF-TWS), and 2-butanol,3-methyl (lowest in MF). Among the 12 aldehydes detected in breads, the most representative were benzaldehyde (0.58–0.79 mg/kg), 2-nonenal (0.33–0.87 mg/kg), and 2-furancarboxaldehyde,5-methyl (0.35–0.99 mg/kg). Semolina breads contained higher concentration of 2-nonenal than mixed flours-based breads (Appendix A). Among alkanes, pentadecane and d-limonene were detected only in semolina and mixed flours-based breads, respectively. Mixed flours-based breads were characterized by presence of benzene compounds. Among the nine ester compounds, ethyl ester of octanoic acid was the most representative (0.78–0.98 mg/kg of bread). Furfural was the most abundant heterocyclic compound (11 compounds), found at the highest (*p* < 0.05) concentration in MF-MFS bread. Among the five organic acids, acetic acid was the most abundant, ranging from 0.10 (S) to 1.11 (MF-MFS) mg/kg of bread, followed by octanoic acid (0.31–0.73 mg/kg). Methyl pyrazine and 2-ethyl-6-methyl pyrazine were the most abundant among the 12 pyrazines detected in bread (Appendix A).

### 3.4. Prediction of Glycaemic Index and Antioxidant Activity of Breads

S-MFS and all the mixed flours-based breads (MF, MF-TWS, and MF-MFS) showed the lowest GI (in vitro) (Table 4).

All the mixed flours-based breads had the highest (*p* < 0.05) in vitro antioxidant activity (Table 4). Both the semolina sourdough breads had higher (*p* < 0.05) antioxidant activity than semolina bread without sourdough (S).

### 3.5. Principal Components Analysis

Based on crumb grain, chemical composition, profiles of FAA and VOCs, PGI and antioxidant activity, breads were differently distributed in the plane formed by the two first principal components, which explained ca. 74% of the total variance of the data (Figure 2). Semolina breads (S-TWS, S-MFS, and S) fell in the left quadrants of the plane, being characterized by higher pH, PGI, and concentrations of some alcohols (phenylethyl alcohol, ethanol, 1-butanol,3-methyl, 3-nonen-1-ol) and 2-nonenal. Higher gas cell areas and antioxidant activity, as well as higher concentrations of fat, lactic acid, D-limonene, benzene compounds, furfural, and 2-ethyl-6-methyl pyrazine characterized the mixed flours-based breads (MF, MF-TWS, and MF-MFS), falling in the right quadrants of the plane. Within these breads, MF-MFS did not cluster with the others, because of its higher concentration of free arginine and tyrosine and lower concentration of acetoin (Figure 2).

Among the highest correlations, fructose was positively correlated with PGI (r = 0.92), 2-acetyl-1-pyrroline with tyrosine (0.91) and phenylalanine (0.90), and pyrazine(1-methyl ethenyl) with proline (0.91). Fructose was negatively correlated with 1h-pyrrole,2-methyl (r = −0.90), ornithine with pyrazine,2,6-dimethyl (r = −0.91), and valine, leucine, and phenylalanine with pyrazine,2-ethenyl-6-methyl (−0.95, −0.93 and −0.97, respectively). In addition, fat was negatively correlated with 2-nonenal (r = −0.95) and 2-nonanone (r = −0.92). Finally, highly negative correlation (r = −0.92) was found between lactic acid and PGI.

### 3.6. Sensory Analysis of Breads

Semolina bread added with traditional sourdough (S-TWS) received the highest score (*p* < 0.05) for crust color (Table 5). The lowest score (*p* < 0.05) for color was found for mixed flours-based bread added with mixed flours sourdough (MF-MFS). The remaining breads received intermediate scores and did not significantly (*p* > 0.05) differ from each other. S bread received the lowest score for acid taste, with significant differences (*p* < 0.05) with respect to all the breads, with the exception of S-TWS. The highest score for aroma was found for semolina sourdough breads (S-TWS and S-MFS), with significant difference (*p* < 0.05) just with respect to mixed flours-based bread without sourdough (MF). The latter did not significantly differ (*p* > 0.05) from the remaining breads (MF-TWS and MF-MFS), in terms of aroma score. Regarding the other attributes evaluated during the panel tests, no difference (*p* > 0.05) was found among the breads. Overall acceptability ranged from ca. 3.0 (MF) to ca. 3.7 (S-TWS and S-MFS breads) (Table 5).

## 4. Discussion

Increasing preference of consumers and bakers towards bread manufactured with mixed flours and/or sourdough [15,18] drove us to investigate about the influence of flours and sourdough on crumb grain, chemical, sensory, and in vitro GI and antioxidant activity of bread. To this aim, we produced and compared six experimental breads: three were produced with a mixture of soft wheat flour, durum wheat semolina, barley, oat, rye, and buckwheat flours; three were semolina-based breads. The variable “sourdough” was considered in both types (mixed flours or semolina) of flour. In addition, two different sourdoughs were used: one (traditional wheat sourdough, TWS) based just on wheat flour, and one based on mixed flours (MFS).

Compared to durum wheat semolina-based breads, those containing soft wheat flour, semolina, barley, oat, rye, and buckwheat flours showed higher specific volume and percentage of gas cell area. This was due to their content (33–37%) in soft wheat flour, well known for its superior bread-making aptitude [2]. In addition, the soft wheat flour used for mixed flours-based bread had a high deformation index (W) value, which is positively correlated to specific volume [31]. Mixed flours-based breads contained higher fat concentration than semolina breads because three (rye, barley, and buckwheat) out of six flours were wholemeal flours, intrinsically richer in fat than refined flours [14]. Fructose and, especially, maltose were the two only fermentable carbohydrates found just in semolina breads. This result could be explained by the high proportion (at least 50%, on flour base) of soft wheat flour (and, conversely, low proportion, around 10%, of semolina) in mixed flours bread. Durum wheat semolina-based sourdoughs were characterized by higher concentrations of fermentable carbohydrates (maltose, glucose, and fructose), compared to soft wheat-based ones [32]. Durum wheat semolina showed 1.6-fold higher maltose concentration than refined (type 00) soft wheat flour [33]. In this study, highest acidification (pH, TTA, and concentration of lactic acid) was found for mixed flours breads added with sourdough as baking improver (MF-TWS and MF-MFS). This could be explained considering the use of sourdough as ingredient, which acted as acidity carrier in bread [34]. However, also mixed flours-based bread without sourdough (MF) showed acidification performances similar to those of MF-TWS and MF-MFS.

Overall, the use of sourdough led, as expected, to increased concentrations of total FAA in bread. Indeed, during sourdough fermentation, proteins are hydrolyzed to peptides and FAA. On the contrary, during dough fermentation by yeasts alone, the latter consume FAA for their growth [35]. Similar results had been found for both experimental [36] and commercial breads [37]. FAA contribute to sensory and nutritional characteristics of baked goods [38]. The mixed flours-based bread added with traditional sourdough (MF-TWS) showed the highest concentration of glutamate, strongly associated to umami taste [39]. Commercial and experimental (started with single strains of *Limosilactobacillus reuteri*) sourdough breads contained higher concentration of glutamate than bread started just with baker’s yeast [40]. The mixed flours-based bread added with mixed flours sourdough (MF-MFS) was rich in alanine, tasting sweet [41] and enhancing the umami taste [42]. In addition, MF-MFS, along with MF-TWS bread, was richest in proline and ornithine. The latter is an amino acid derivative, often used as dietary supplement putatively having antifatigue, muscle synthesis enhancing, basal metabolism accelerating [43], wrinkle reducing [44], and immune-modulating [45] effects. Diana et al. [37] reported that an experimental sourdough bread, started with a selected strain of *Levilactobacillus brevis*, contained ornithine at a slightly lower concentration than our mixed flours-based sourdough breads. Remarkably, our MF-MFS bread was rich in some essential FAA: lysine, leucine, isoleucine, valine, phenylalanine, tyrosine, and tryptophan. In addition, MF-MFS bread contained the highest concentration of the amino acid derivative GABA, with antihypertensive activity and neuro-active function. Previous research [37,46] found that experimental breads contained concentrations of GABA similar to those found in this study. Ingestion of a beverage supplemented with 50 mg of GABA decreased fatigue and increased task-solving ability in 30 healthy subjects [47]. We hypothesize that ingestion of ca. 180 g of MF-MFS bread, a little bit higher than the average daily per capita consumption in Europe [17], would be sufficient to benefit brain performance. In addition, 180 g of that bread would provide, considering just FAA, all (referring to isoleucine, phenylalanine, tyrosine, and tryptophan) or majority (66%, 57% and 97% for lysine, leucine, and valine, respectively) of the amino acids daily intake requirement of an adult weighing 60 kg [48].

Sensory features of breads were assessed through determination of VOCs profile and panel test. Type of flour affected VOCs profile of bread. Indeed, semolina breads were characterized by 2-nonenal (fatty, green odor) and phenylethyl alcohol (flowery, yeast-like), which was already recognized as important odor compounds of wheat bread crumb [49]. High concentrations of ethanol and 3-methyl butanol (synonym: isoamyl alcohol), having balsamic, alcoholic, and malty notes, also distinguished semolina breads from mixed flours-based breads. Mixed flours-based breads were characterized by higher concentration of furfural (sweet, woody, almond odor) than semolina breads and by the presence of d-limonene (citrus). Van Kerrebroeck et al. [50] had shown that use of teff, instead of wheat flour, during sourdough fermentation impacted on VOCs profile of the resulting bread.

Overall, compared to type of flours, the addition of sourdough had lower influence on VOCs profiles of bread. We may hypothesize that this result was due to both the low percentage (9.8%) of sourdough used and the limited (1.5 h) fermentation time after sourdough addition. As regards semolina breads, very few compounds (e.g., 2-butanol,3-methyl; 2-furancarboxyaldheyde,5-methyl; benzeneacetaldheyde; acetic acid) increased, possibly due to addition of sourdough. In these breads, 1-butanol,3-methyl was even lower than the relative bread without sourdough addition. This result was in disagreement with previous findings, showing that sourdough breads contained higher concentrations of higher alcohols (including 1-butanol,3-methyl) than breads started just with baker’s yeast [50,51]. The reason for that could be the use of a relatively long (16 h) pre-fermentation for all the breads, including those without addition of sourdough. We hypothesize that this pre-fermentation step could have impacted on levels of some VOCs more than the final short (1.5 h) fermentation. Concerning mixed flours-based breads, acetic acid (putatively produced by both lactic acid bacteria, via heterolactic fermentation, and yeasts, as minor product of alcoholic fermentation) was the only VOC found at higher concentration when sourdough had been used, compared to MF (bread without sourdough). This result confirms the role of sourdough as acidifying ingredient [50]. Furthermore, comparing mixed flours-based bread added with mixed flours sourdough (MF-MFS) to MF, we found that some VOCs (e.g., 2-butanol,3-methyl; 2-furancarboxyaldheyde,5-methyl; furfural; benzeneacetaldehyde) increased. Overall, the use of mixed flours-based sourdough (MFS) led to quite a distinct profile of VOCs, mainly because of highest concentrations of furfural, d-limonene, and acetic acid and lowest concentration of acetoin (butter odor). We may hypothesize that this result could be due to differences between microbial communities harbored in the two sourdoughs (MFS and TWS) used as ingredient. The influence of the type of sourdough on the concentration of VOCs in bread had been previously reported [51,52].

Results from the panel test performed on the breads object of study only partially reflected the different VOCs profiles, with lower score for aroma of mixed flours-based bread without sourdough (MF), compared to both semolina-based sourdough breads (S-TWS and S-MFS). In addition, the lowest score for acid taste received by the semolina bread without sourdough (S) corresponded to the lowest concentration of acetic acid. Overall, all the breads resulted acceptable.

Prediction of GI showed that all the mixed flours-based breads had lower GI than semolina breads and could be regarded as moderate GI foods [53]. This result may be explained by the use of flours that contained one or more compounds capable of lowering GI. In detail, buckwheat contains flavonoids that inhibit α-amylase, which, in turn, lowers the digestion rate of starch [54]. Oat and barley are rich in soluble dietary fiber (e.g., β-glucan), which decreases the rate at which glucose is released to intestine and blood [55]. In addition, barley contains d-fagomine, an iminosugar that would slow down the release of postprandial glucose from starch and other carbohydrates [56]. We hypothesize that the relatively low GI predicted in this study for mixed flours-based breads could be due to a combination of variables, such as percentage of amylose, interactions between starch and other components, concentration of organic acids, and particle size [1]. Among the variables subjected to analyses, absence of maltose and higher acidification in mixed-flours breads, compared to semolina breads, could have contributed to decrease GI. The use of sourdough had much lower impact on in vitro GI than type of flours used for breadmaking. This result contrasts to the very well-known GI-lowering activity of sourdough bread [16].

Prediction of antioxidant activity showed that mixed flours-based breads had the highest values of radical scavenging in vitro activity. Again, this may be attributable to the flours used in these breads to replace partially wheat flour. These flours contain antioxidant molecules, such as phenolic compounds (e.g., ferulic acid) [57], and, in case of oat flour, the alkaloid avenanthramides [58]. Previous studies had shown in vitro and/or in vivo antioxidant activity of breads supplemented with barley [59], buckwheat [60] flours, as well as of breads totally made of rye flour [61]. Under our experimental conditions, the use of sourdough had limited (in case of semolina breads) or no (in case of mixed-flours breads) effect on antioxidant activity. This result is in disagreement with current literature, which highlights that concentration of extractable phenolic compounds in flour increases during sourdough fermentation, thus enhancing antioxidant activity of leavened baked goods [16]. We hypothesize that the result found in this study was much affected by the low percentage (9.8%) of sourdough in bread recipe, as well as by the short time (1.5 h) passed from the addition of sourdough to baking. It is probable that if we had used either higher percentage of sourdough, or longer sourdough fermentation, antioxidant activity of bread would have significantly increased.

In this study, based on the PCA built on crumb grain, chemical composition, profiles of FAA and VOCs, and prediction of GI and antioxidant activity of breads, the use of mixed flours mostly contributed to differentiate breads, in terms of some VOCs, lactic acid, in vitro GI, and antioxidant activity. Yet, the use of sourdough, especially of that based on mixed flours (MFS), led to increased concentration of some FAA. Several correlations were observed between some FAA and VOCs (pyrrolines and pyrazines) typically generated during baking, upon Maillard reactions (involving sugars as well) [62]. Negative correlations could be explained considering that FAA degradation might induce formation of pyrazines and pyrroles, among others [63]. Positive correlations between FAA and products of Maillard reactions are difficult to be interpreted. However, in a recent study about hazelnuts, alanine was positively correlated with 2,5-dimethyl pyrazine [64]. We found that fat was negatively correlated with C9-aldehyde and -ketone. It is possible that these two VOCs derived from lipid oxidation, regarded as the second (after fermentation sensu latu) most important pathway contributing to aroma of bread crumb [65]. GI was positively and negatively correlated with fructose and lactic acid, respectively. It is well known that organic acids reduce starch digestibility and this, in turn, lowers GI of bread [16].

## 5. Conclusions

Partial replacement of wheat with barley, oat, rye, and buckwheat flours gave breads with lower GI and higher antioxidant activity than semolina breads. All the mixed flours-based bread, although showing VOCs profiles differing from those of semolina breads, resulted acceptable. In addition, the use of a mixed flours sourdough allowed obtaining a bread (MF-MFS) that, besides being a food with moderate GI and antioxidant activity, was characterized by high concentrations of essential FAA and GABA. Overall, the use of sourdough drove to breads with higher concentration of total and individual FAA than breads without sourdough addition. On the contrary, it affected very little VOCs profile and in vitro GI and antioxidant activity of breads.

## Figures and Tables

**Figure 1 foods-10-01613-f001:**
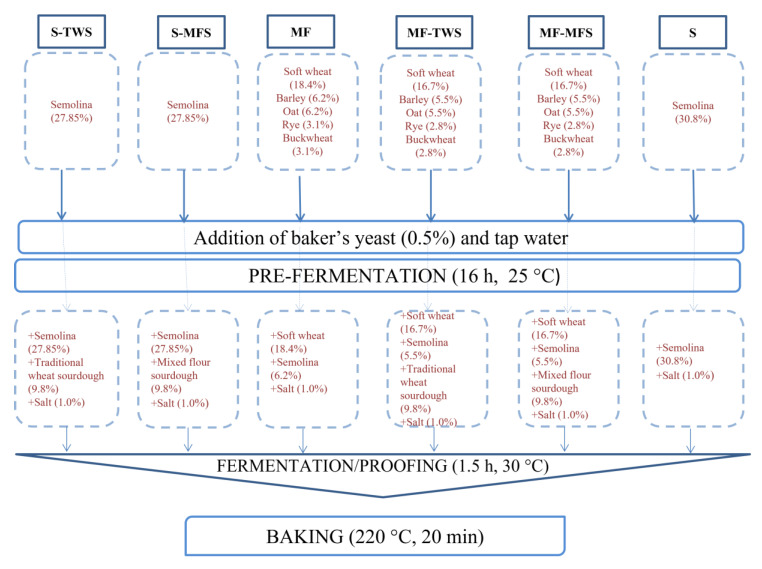
Protocol adopted for manufacturing S-TWS, S-MFS, MF, MF-TWS, MF-MFS, and S breads.

**Figure 2 foods-10-01613-f002:**
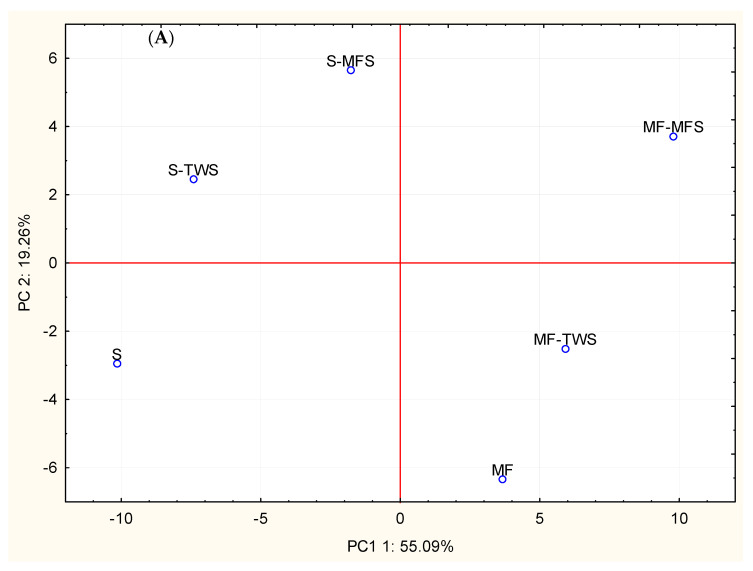
Score (**A**) and loading (**B**) plots of the first and second principal components after Principal Component Analysis based on crumb grain, chemical composition, profiles of FAA and VOCs, PGI and antioxidant activity of S-TWS, S-MFS, MF, MF-TWS, MF-MFS, and S breads.

**Table 1 foods-10-01613-t001:** Formulae of bread doughs. Ingredients are expressed as percentage of dough weight.

	Soft Wheat	Semolina	Barley	Oat	Rye	Buckwheat	Water	Sourdough
S	/	61.6	/	/	/	/	36.9	/
S-TWS	/	55.7	/	/	/	/	33.0	9.8
S-MFS	/	55.7	/	/	/	/	33.0	9.8
MF	36.8	6.2	6.2	6.2	3.1	3.1	36.9	/
MF-TWS	33.4	5.5	5.5	5.5	2.8	2.8	33.2	9.8
MF-MFS	33.4	5.5	5.5	5.5	2.8	2.8	33.2	9.8

**Table 2 foods-10-01613-t002:** Crumb grain and chemical characteristics of S-TWS, S-MFS, MF, MF-TWS, MF-MFS, and S breads.

	Semolina with Traditional Sourdough (S-TWS)	Semolina with Mixed Flours Sourdough (S-MFS)	Mixed Flours (MF)	Mixed Flours with Traditional Sourdough (MF-TWS)	Mixed Flours with Mixed Flours Sourdough (MF-MFS)	Semolina (S)
Specific volume (cm^3^/g)	2.9 ± 0.10 ^b^	2.8 ± 0.16 ^b^	3.3 ± 0.13 ^a^	3.2 ± 0.15 ^a^	3.2 ± 0.10 ^a^	2.9 ± 0.11 ^b^
Gas cell area (%)	33.2 ± 1.70 ^d^	37.0 ± 1.77 ^c^	56.7 ± 2.85 ^b^	52.3 ± 2.60 ^b^	61.7 ± 3.10 ^a^	30.6 ± 1.53 ^d^
Ash (%)	1.75 ± 0.035 ^a^	1.77 ± 0.030 ^a^	1.74 ± 0.026 ^a^	1.75 ± 0.024 ^a^	1.70 ± 0.034 ^a^	1.71 ± 0.020 ^a^
Moisture (%)	32 ± 1.3 ^a^	33 ± 1.0 ^a^	33 ± 0.9 ^a^	32 ± 1.2 ^a^	33 ± 1.3 ^a^	32 ± 1.0 ^a^
Fat (%)	1.1 ± 0.13 ^b^	1.2 ± 0.15 ^b^	1.6 ± 0.19 ^a^	1.7 ± 0.25 ^a^	1.8 ± 0.21 ^a^	1.1 ± 0.10 ^b^
Total saturated fatty acids (%)	0.21 ± 0.100 ^a^	0.22 ± 0.140 ^a^	0.32 ± 0.110 ^a^	0.30 ± 0.120 ^a^	0.32 ± 0.140 ^a^	0.22 ± 0.130 ^a^
Total carbohydrates (%)	52 ± 2.8 ^a^	51 ± 3.0 ^a^	52 ± 2.5 ^a^	52 ± 2.4 ^a^	50 ± 2.0 ^a^	50 ± 2.5 ^a^
Fructose (%)	0.12 ± 0.020 ^a^	0.10 ± 0.022 ^a^	0.00 ^b^	0.00 ^b^	0.00 ^b^	0.12 ± 0.019 ^a^
Maltose (%)	0.52 ± 0.022 ^a^	0.51 ± 0.025 ^a^	0.00 ^b^	0.00 ^b^	0.00 ^b^	0.52 ± 0.023 ^a^
Proteins (%)	10.4 ± 0.75 ^a^	10.4 ± 0.68 ^a^	10.5 ± 0.70 ^a^	10.6 ± 0.75 ^a^	10.7 ± 0.66 ^a^	10.6 ± 0.69 ^a^
Salt (%)	1.3 ± 0.19 ^a^	1.2 ± 0.15 ^a^	1.3 ± 0.20 ^a^	1.1 ± 0.18 ^a^	1.2 ± 0.19 ^a^	1.3 ± 0.18 ^a^
Soluble dietary fiber (%)	1.11 ± 0.54 ^a^	1.09 ± 0.47 ^a^	1.11 ± 0.50 ^a^	1.10 ± 0.55 ^a^	1.09 ± 0.43 ^a^	1.09 ± 0.50 ^a^
Total dietary fiber (%)	3.8 ± 0.60 ^a^	3.8 ± 0.50 ^a^	3.8 ± 0.62 ^a^	3.9 ± 0.59 ^a^	3.9 ± 0.57 ^a^	3.8 ± 0.55 ^a^
pH	5.2 ± 0.06 ^b^	5.1 ± 0.05 ^b^	5.0 ± 0.02 ^c^	4.9 ± 0.02 ^d^	4.9 ± 0.03 ^d^	5.6 ± 0.04 ^a^
TTA (ml of NaOH 0.1 M)	1.4 ± 0.10 ^c^	1.3 ± 0.08 ^c^	1.9 ± 0.04 ^b^	2.0 ± 0.03 ^a^	2.0 ± 0.03 ^a^	0.9 ± 0.04 ^d^
Lactic acid (mmol/kg)	13 ± 0.7 ^b^	14 ± 0.8 ^b^	19 ± 0.7 ^a^	19 ± 0.6 ^a^	20 ± 0.7 ^a^	10 ± 0.5 ^c^
Acetic acid (mmol/kg)	3 ± 0.4 ^b^	3 ± 0.5 ^b^	0.00 ^c^	3 ± 0.3 ^b^	6 ± 0.4 ^a^	0.00 ^c^

Values (mean of three independent experiments analyzed in duplicate ± standard deviation) in the same row with different superscript letters are significantly different (*p* < 0.05).

**Table 3 foods-10-01613-t003:** Concentration (mg/kg) of individual and total (total FAA) free amino acids and their derivatives (GABA, ornithine) in S-TWS, S-MFS, MF, MF-TWS, MF-MFS, and S breads.

	S-TWS	S-MFS	MF	MF-TWS	MF-MFS	S
asp	473 ± 14 ^d^	586 ± 15 ^b^	286 ± 20 ^e^	519 ± 16 ^c^	612 ± 24 ^a^	200 ± 13 ^f^
thr	66 ± 9 ^ab^	54 ± 7 ^c^	71 ± 8 ^a^	54 ± 5 ^c^	60 ± 6 ^bc^	60 ± 7 ^bc^
ser	47 ± 6 ^c^	58 ± 8 ^b^	47 ± 5 ^c^	74 ± 10 ^a^	37 ± 5 ^d^	37 ± 6 ^d^
glu	1008 ± 22 ^b^	890 ± 10 ^c^	706 ± 17 ^d^	1104 ± 34 ^a^	1023 ± 19 ^b^	537 ± 15 ^e^
gly	105 ± 10 ^bc^	109 ± 6 ^b^	94 ± 8 ^c^	146 ± 11 ^a^	139 ± 8 ^a^	56 ± 6 ^d^
ala	285 ± 5 ^d^	343 ± 12 ^b^	169 ± 8 ^e^	307 ± 10 ^c^	370 ± 7 ^a^	147 ± 5 ^f^
cys	72 ± 6 ^c^	84 ± 8 ^b^	84 ± 5 ^b^	96 ± 6 ^a^	84 ± 6 ^b^	72 ± 7 ^c^
val	59 ± 7 ^d^	123 ± 9 ^b^	35 ± 9 ^e^	94 ± 10 ^c^	141 ± 10 ^a^	23 ± 10 ^e^
met	82 ± 8 ^a^	75 ± 8 ^ab^	52 ± 6 ^c^	67 ± 7 ^b^	67 ± 9 ^b^	75 ± 9 ^ab^
ile	0 ^b^	0 ^b^	0 ^b^	157 ± 10 ^a^	157 ± 7 ^a^	0 ^b^
leu	0 ^d^	72 ± 8 ^c^	0 ^d^	98 ± 7 ^b^	125 ± 8 ^a^	0 ^d^
tyr	63 ± 5 ^b^	100 ± 6 ^a^	27 ± 7 ^c^	63 ± 7 ^b^	100 ± 8 ^a^	27 ± 5 ^c^
phe	33 ± 8 ^d^	91 ± 7 ^b^	25 ± 8 ^ef^	58 ± 10 ^c^	116 ± 8 ^a^	17 ± 8 ^f^
GABA	227 ± 16 ^cd^	211 ± 14 ^d^	134 ± 9 ^e^	242 ± 18 ^bc^	278 ± 12 ^a^	113 ± 8 ^f^
his	23 ± 8 ^a^	0 ^b^	0 ^b^	0 ^b^	31 ± 7 ^a^	0 ^b^
trp	0 ^b^	0 ^b^	0 ^b^	0 ^b^	41 ± 6 ^a^	0 ^b^
orn	59 ± 7 ^c^	33 ± 7 ^d^	99 ± 6 ^b^	205 ± 13 ^a^	192 ± 15 ^a^	26 ± 7 ^d^
lys	80 ± 6 ^b^	80 ± 7 ^b^	58 ± 5 ^c^	110 ± 9 ^a^	110 ± 12 ^a^	44 ± 8 ^d^
arg	105 ± 9 ^c^	226 ± 10 ^a^	0 ^d^	113 ± 8 ^c^	174 ± 13 ^b^	0 ^d^
pro	184 ± 9 ^b^	150 ± 6 ^c^	178 ± 7 ^b^	276 ± 12 ^a^	276 ± 16 ^a^	92 ± 5 ^d^
total FAA	2971 ± 155 ^d^	3284 ± 148 ^c^	2068 ± 128 ^e^	3784 ± 203 ^b^	4131 ± 206 ^a^	1526 ± 121 ^f^

Values (mean of three independent experiments analyzed in duplicate ± standard deviation) in the same row with different superscript letters are significantly different (*p* < 0.05).

**Table 4 foods-10-01613-t004:** Prediction of glycaemic index (PGI) and antioxidant activity (expressed as percentage of DPPH• scavenging activity) of S-TWS, S-MFS, MF, MF-TWS, MF-MFS, and S breads.

	S-TWS	S-MFS	MF	MF-TWS	MF-MFS	S
PGI (%)	80 ± 3 ^a^	72 ± 3 ^b^	68 ± 4 ^b^	68 ± 4 ^b^	68 ± 5 ^b^	79 ± 4 ^a^
DPPH• scavenging activity (%)	48.0 ± 1.80 ^b^	46.0 ± 2.04 ^b^	58.5 ± 2.20 ^a^	61.0 ± 2.51 ^a^	63.5 ± 2.02 ^a^	42.0 ± 2.10 ^c^

Values (mean of three independent experiments analyzed in duplicate ± standard deviation) in the same row with different superscript letters are significantly different (*p* < 0.05).

**Table 5 foods-10-01613-t005:** Average scores for the attributes used during the panel test carried out on S-TWS, S-MFS, MF, MF-TWS, MF-MFS, and S breads.

	S-TWS	S-MFS	MF	MF-TWS	MF-MFS	S
Crust color	3.7 ± 0.49 ^a^	2.9 ± 0.90 ^b^	3.0 ± 0.58 ^b^	2.7 ± 0.76 ^b^	2.3 ± 0.76 ^c^	2.6 ± 0.53 ^b^
Odor	3.4 ± 0.79 ^a^	3.0 ± 0.82 ^a^	2.9 ± 0.69 ^a^	2.9 ± 0.69 ^a^	2.9 ± 0.69 ^a^	3.6 ± 0.98 ^a^
Crispness	3.4 ± 1.13 ^a^	3.7 ± 0.95 ^a^	3.0 ± 0.82 ^a^	3.1 ± 1.21 ^a^	2.9 ± 1.21 ^a^	2.9 ± 1.21 ^a^
Elasticity	3.7 ± 0.76 ^a^	3.3 ± 0.95 ^a^	3.7 ± 0.76 ^a^	3.3 ± 1.38 ^a^	3.3 ± 1.25 ^a^	3.1 ± 1.21 ^a^
Masticability	4.0 ± 0.58 ^a^	3.9 ± 0.38 ^a^	3.4 ± 0.79 ^a^	3.4 ± 0.98 ^a^	3.6 ± 0.79 ^a^	3.9 ± 0.38 ^a^
Acid taste	2.0 ± 1.15 ^ab^	2.4 ± 0.98 ^a^	2.1 ± 0.90 ^a^	2.6 ± 1.27 ^a^	2.9 ± 1.07 ^a^	1.1 ± 0.90 ^b^
Sweet taste	3.1 ± 0.90 ^a^	2.9 ± 0.90 ^a^	2.6 ± 1.27 ^a^	2.6 ± 1.27 ^a^	2.4 ± 1.27 ^a^	2.9 ± 0.69 ^a^
Salty taste	2.7 ± 1.11 ^a^	2.6 ± 1.27 ^a^	2.4 ± 1.27 ^a^	2.4 ± 1.51 ^a^	2.9 ± 1.57 ^a^	2.4 ± 0.79 ^a^
Aroma	3.7 ± 0.76 ^a^	3.6 ± 0.79 ^a^	2.7 ± 0.76 ^b^	3.0 ± 1.15 ^ab^	2.9 ± 1.07 ^ab^	3.4 ± 0.98 ^ab^
Overall acceptability	3.7 ± 0.76 ^a^	3.7 ± 0.76 ^a^	3.0 ± 0.82 ^a^	3.1 ± 0.90 ^a^	3.1 ± 1.07 ^a^	3.0 ± 0.82 ^a^

Values (mean of three independent experiments analyzed in duplicate ± standard deviation) in the same row with different superscript letters are significantly different (*p* < 0.05).

## Data Availability

Data is contained within this article and related Appendix A.

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
