# Peer review of "Functional, Nutritional, and Sensory Quality of Mixed Flours-Based Breads as Compared to Durum Wheat Semolina-Based Breads"

_foods, 2021, doi:10.3390/foods10071613_

Round 1

Reviewer 1 Report

Dear Authors,

The manuscript in a present from sound better, although there is a need to introduce few explanations:

l. 498-500  This result was in disagreement with previous findings, showing that sourdough breads contained higher concentrations of higher alcohols (including 1-butanol,3-methyl) than breads started just with baker’s yeast' - please explain the potential reason of such phanomenon

l. 544-546 'This result is in disagreement with current literature, which highlights that concentration of extractable phenolic compounds in flour in creases during sourdough fermentation, thus enhancing antioxidant activity of leavened baked goods' - please explain the potential reason of such phanomenon

l. 574-576 'Although these results could sound odd in the frame of the abundant literature about sensorial, functional and nutritional effects of sourdough, it has to be taken into account that we used just 9.8% of sourdough in bread recipe and that just 1.5 h passed from the addition of sourdough to baking.' - please reformulate (remove 'could sound odd, etc) and move that sentence to the discussion section.

Author Response

  1. 498-500 This result was in disagreement with previous findings, showing that sourdough breads contained higher concentrations of higher alcohols (including 1-butanol,3-methyl) than breads started just with baker’s yeast' - please explain the potential reason of such phanomenon
  • AU: OK, we have explained a possible reason for such phenomenon (ll. 505-508).
  1. 544-546 'This result is in disagreement with current literature, which highlights that concentration of extractable phenolic compounds in flour in creases during sourdough fermentation, thus enhancing antioxidant activity of leavened baked goods' - please explain the potential reason of such phanomenon
  • AU: OK, we have explained a possible reason for such phenomenon (ll. 555-557).
  1. 574-576 'Although these results could sound odd in the frame of the abundant literature about sensorial, functional and nutritional effects of sourdough, it has to be taken into account that we used just 9.8% of sourdough in bread recipe and that just 1.5 h passed from the addition of sourdough to baking.' - please reformulate (remove 'could sound odd, etc) and move that sentence to the discussion section.
  • AU: OK, the sentence has been reformulated and moved to the discussion (ll. 555-559).

Reviewer 2 Report

Thank you for sending me a revised version of the manuscript. In my opinion, the paper was significantly improved and all my comments were addressed. However, there are still some little points to take into account (as follows).

Line 183-184: I guess it is a comment to reviewers. “We used baker’s yeast also for bread added with sourdough because it guarantees dough 184 leavenings in a short time (during the second step).” I think it shouldn’t be included in the manuscript in this form.

Conclusions: “Overall, the use of sourdough profoundly impacted on the concentration of total and individual FFA.” – please be more precise in making conclusions.

Author Response

Line 183-184: I guess it is a comment to reviewers. “We used baker’s yeast also for bread added with sourdough because it guarantees dough 184 leavenings in a short time (during the second step).” I think it shouldn’t be included in the manuscript in this form.

  • AU: OK, in the original manuscript this sentence was missing. However, during the first revision, the reviewer 1 explicitly asked to “explain why did you use baker’s yeast for all breads (even with sourdough)”. Therefore, those lines (ll. 141-142) have to be left; they are not a comment to reviewers.

Conclusions: “Overall, the use of sourdough profoundly impacted on the concentration of total and individual FFA.” – please be more precise in making conclusions.

  • AU: OK, we have revised this sentence, according to the suggestion from the reviewer (ll. 587-588).

Reviewer 3 Report

The authors addressed adequately the reviewers' suggestions, but it needs to be carefully revised about English language.

Author Response

The authors addressed adequately the reviewers' suggestions, but it needs to be carefully revised about English language.

  • AU: OK, while revising the manuscript, we have paid attention to any misspelling and grammar mistakes we could find.